# Traumatic dental injuries in permanent teeth among Arab children: prevalence, and associated risk factors—a systematic review and meta-analysis

Raghad Hashim[1], Alexander Maniangat Luke[2], Afraa Salah[3] and Simy Mathew[4]

[1] Department of Basic Medical and Dental Sciences, Centre of Medical and Bio-allied Health Sciences Research, Ajman University, Ajman, United Arab Emirates
[2] Department of Clinical Sciences, Centre of Medical and Bio-allied Health Sciences Research, Ajman University, Ajman, United Arab Emirates
[3] Department of Clinical Sciences, Ajman University, Ajman, United Arab Emirates
[4] Institute of Public Health, College of Medicine & Health Sciences, UAE University, Al Ain, United Arab Emirates

Corresponding author
Alexander Maniangat Luke,
a.luke@ajman.ac.ae

## ABSTRACT

**Introduction:** There is no clear literature present till date assessing the prevalence of traumatic dental injuries and associated factors in children living in Arab countries. The purpose of this study was to systematically assess the prevalence, trends, and potential risk factors of traumatic dental injury (TDI) in permanent teeth among children and adolescents in Arab countries.

**Methods:** This study followed the Preferred Reporting Items for Systematic Reviews and Meta-Analyses (PRISMA) guidelines. The researchers conducted a comprehensive literature search in various databases, including PubMed, Cochrane library of systematic reviews, Directory of Open Access Journals (DOAJ), Google Scholar, and gray literature sources such as MDS dissertations and manuscripts. To analyze the collected data, the researchers used a random effects model for conducting meta-analyses. Pooled estimates of prevalence and odds ratios were derived, along with 95% confidence intervals (CI), to provide a measure of statistical precision and variability in the findings. For the studies that were considered, trend analysis was done. The risk of bias assessment of included studies was done using Newcastle–Ottawa tool for cross-sectional studies.

**Results:** A total of 545 articles were identified, of which 23 articles fulfilled the inclusion criteria. Quality assessment of included studies showed that twenty studies were of high quality and three studies showed low quality. The frequency of dental trauma in Arab children was 26% (95% CI [10–43]). Children with overjet of more than 3.5 mm were shown to be 1.78 times more likely to have dental injury than children with normal overjet (pooled odds ratio 1.78; 95% CI [1.17–2.70]). Males had a 2.06 times odds of TDI compared to females. Children with insufficient lip coverage had an odds ratio of 2.57, indicating a higher likelihood of oral injuries compared to those with appropriate lip coverage.

**Conclusion:** Significant correlations were obtained between the prevalence of dental trauma and other variables such as male gender, increased overjet, inadequate lip coverage, *etc*. Future population-based analytical research should concentrate on

documenting the incidence and/or prevalence of TDI among marginalized communities in order to better understand the primary causes of TDI.

# INTRODUCTION

Children are highly susceptible to traumatic dental injury (TDI), which is well-recognized as a serious problem in public health (*Azami-Aghdash et al., 2015*; *Tello et al., 2016*). Injuries to permanent teeth are a common cause of pain and discomfort in the craniofacial area (*Andreasen, Andreasen & Andersson, 2018*). Traumatic dental injuries can occur to anyone at any time, with an incidence that ranges from 4% to 58% (*Abdel Malak et al., 2021*). To pinpoint the likely risk factors linked to TDI, numerous research have been carried out. Being of male gender (*Dharmani, Pathak & Sidhu, 2019*; *Ogordi, Ize-Iyamu & Adeniyi, 2019*; *Ain et al., 2016*), an overjet of more than 5 mm at the incisors, inadequate lip coverage (*Garg et al., 2017*; *Al Suwyed et al., 2021*), low socioeconomic status, and the presence of severe dental injuries (*Lisboa et al., 2013*) have all been determined to be significant risk factors. The results indicated that compared to children whose TDIs were successfully treated, those whose TDIs were left untreated were 20 times more likely to report that their condition negatively affected their QoL (*Lopez et al., 2019*). Children who suffer from TDIs may need hospitalization, and some of these injuries might have lasting effects on their growth and development (*Albedewi et al., 2021*). Children and adolescents make up about 17.7% of the population in the UAE (*Puri-Mirza, 2020*). Due to the fact that these injuries are most common in adolescence, these injuries among teenagers are particularly concerning (*Petti, Glendor & Andersson, 2018*). In Arab countries, there is a lack of data on the prevalence of dental injuries among children. The development of strategies to mitigate the deleterious effects of oral trauma might benefit from a better understanding of some of the characteristics associated with these injuries.

Data from the primary studies show substantial variation and heterogeneity (*Oldin et al., 2016*; *Magno et al., 2020*; *Abdel Malak et al., 2021*). Unfortunately, there is a lack of hard data about long-term tendencies and other crucial criteria, such as socioeconomic status and obesity, for TDI Arab countries. These significant gaps hinder careful planning, improved judgment, and effective intervention design, all of which are necessary for the creation of a successful intervention for the treatment and prevention of TDI.

This study aimed to generate a reliable pooled estimate by conducting a comprehensive review and meta-analysis of the existing literature to assess long-term trends, the prevalence of TDI in permanent dentition, and the detectable correlations between important risk variables in Arab children and adolescents.

A systematic review and meta-analysis of dental trauma in the Arab population may reveal gaps in the literature such as:

**Research is limited**: It is probable that there is a lack of original studies on oral trauma in the Arab population in the published literature. Lack of knowledge, limited funding, or cultural factors affecting research priorities might all be to blame.

**Heterogeneity of methodology**: It may be difficult to compare and combine the results of the existing research because they may have used different methodologies, diagnostic standards, and sampling procedures. The validity and dependability of the meta-analysis may be impacted by this heterogeneity.

**Geographic representation:** The occurrence of dental trauma may range between various Arab nations or regions. The generalizability of the results to the entire Arab population may be constrained by the studies' insufficient representation of certain Arab countries or areas.

**Language bias:** Studies published only in certain languages, such as English, may be included, leaving out pertinent studies published in other languages. The review's thoroughness and representativeness may suffer as a result of this prejudice.

**Disparities by gender and age:** Dental trauma may occur at a different rate across different age groups and sexes. If the existing research does not adequately represent the diverse age ranges and gender distribution within the Arab population, prevalence figures may be erroneous and distorted.

**Lack of standardized reporting:** In the known research, there are no standardized reporting processes or guidelines, which could result in inaccurate or inconsistent data reporting. As a result, it may be difficult to accurately collect and synthesize data throughout the systematic review process.

A thorough and methodical strategy is needed to fill in these research gaps. To find a wider variety of research, it entails undertaking a thorough literature search across numerous databases, including both English and non-English sources. Additionally, efforts can be made to persuade researchers to adopt the standardized methodology and reporting practices and to perform studies that especially target oral trauma in the Arab community.

The prevalence of oral trauma in the Arab population can be better understood by solving these research gaps, which can also help to inform clinical practice, guide policy decisions, and identify areas that require further research.

## METHODS

**Protocol and registration:** The PRISMA criteria for reporting systematic reviews served as the foundation for this reporting. Before the review began, the review protocol was registered with PROSPERO (ID number CRD42023421734). This study is a systematic review and meta-analysis of previously published data and did not require ethical approval, as no human subjects were used in the preparation of this manuscript.

### Eligibility criteria

The Population, Exposure, Outcome, and Study Design (PEOS) structure for cross-sectional research was utilized to develop the focused review question. The population consisted of children between 6 to 17 years from Arab nations, the exposure

was damage to their permanent teeth, the outcome was the frequency of the dental injuries that occurred, and the study designs were cross-sectional in nature.

The eligibility criteria was included studies with human participants with permanent dentition in age group 6–17 years suffering from trauma to anterior or posterior teeth due to any reason. Studies published until June 2023 were included. Studies that did not fall within PICO were excluded such as population with dental trauma other than Arab children, trauma to deciduous dentition, *in vitro* studies, *etc.*

**Search technique:** In May 2023, a sophisticated, methodical search was carried out across various databases, including PubMed, MEDLINE, PubMed Central and the Directory of Open Access Journals (DOAJ). Medical Subject Headings (MeSH) and non-MeSH terminology were taken into consideration when choosing the keywords. In the advanced search option, these keywords and MeSH phrases were combined with Boolean operators.

## STRATEGY

**Population:** Adolescent [MeSH] OR OR Teen[Text Word] OR Young teen[Text Word] OR Child [Text Word] Population or Schoolchildren [Text] OR Children [Text] Children [Text] OR Children [Text] OR Children [Text] OR Youth [Text].

**Exposure:** (((trauma"[Subheading] OR (((injuries"[All Fields] OR (((wounds and injuries"[MeSH Terms] OR (((tooth fractures"[MeSH Terms] OR ((tooth"[All Fields] AND" fractures "[All Fields] OR ((tooth"[All Fields] AND "dental fracture"[All Fields])) AND ((dentition, permanent"[MeSH Terms] OR (permanent"[All Fields] AND "dentition"[All Fields] OR (permanent"[All Fields] AND" teeth "[All Fields] OR (permanent"[All Fields] AND "teeth")).

**Outcome:** "epidemiology" [Subtitle] EITHER "epidemiology" [Everything Inside the field] OR "prevalence"[MeSH Terms] OR "prevalence" [Everything Inside the field].

**Study designs** include prospective, cross-sectional, and observational studies. The search terms: #1 AND #2 AND #3 AND #4

**Database search:** Electronic databases such as MEDLINE, DOAJ, Cochrane, PubMeD Central.

**Grey Literature:** SIGLE (opengrey.eu) and greylit.org.

**Internet search engines:** PubMed and Google Scholar.

### Study selection

Two reviewers (S.M and A.S) separately evaluated the titles and abstracts of the articles they had found using the search approach to find studies that might be eligible. Any conflicts that arose were discussed with a third reviewer (R.H or A.M.L). However, there were no disputes identified between two reviewers for study selection. The selection criteria for the studies included full-text, English-language academic articles. Case reports, case series, narrative reviews, one-arm longitudinal studies, and *in vitro* studies were excluded and all cross-sectional and observational research were included. The present systematic review did not perform or report on any abstracts lacking full-text articles. Any studies reporting information on trauma to deciduous teeth were also excluded.
## Extraction of data

Duplicate articles were removed with the help of EndNote reference manager software (EndNote X9) (*The EndNote Team, 2013*). Manual checking of duplicate articles was also done independently by two reviewers (S.M and A.S). The study investigators went through the articles found after doing the literature search to weed out duplicates and irrelevant studies. Two reviewers (A.M.L and R.H) separately conducted additional screening of the remaining articles to determine whether they were eligible to include within the scope of the evaluation. Data on the study design, location of the study, population, age, gender, prevalence of dental trauma, children's residential areas (rural *vs.* urban), overjet, lip competence, the reason for trauma (fall, violence, sports, biting, *etc.*), and type of dental trauma were taken from the final set of articles (teeth fracture, enamel, and dentin cracking, missing due to trauma, *etc*).

## Evaluation of potential for bias

Each study's potential for bias was evaluated by two reviewers, using the Newcastle–Ottawa scale modified for cross-sectional research (*Moskalewicz & Oremus, 2020*) based on five factors: (1) sample representativeness (all subjects included or random sampling used); (2) sample size (supported using techniques like strength assessment); (3) inter-group comparability; (4) proven oral trauma screening protocols; and (5) sufficiency of descriptive statistics. The overall quality rating varied from 0 to 5. Scores of 3 and 3 were used to denote high- and low-quality scores, respectively. Potential differences of opinion with an impartial third author were consulted on data extraction and quality assessment.

## Synthesis and evaluation of data

MetaXL a plug-in for Microsoft Excel (*Microsoft Corporation, 2018*) designed specifically for meta-analysis, was used for all statistical calculations (www.epigear.com). The QE model was used to synthesize the pooled prevalence based on the quality score given to each research. The quality of each study was quantified by dividing its quality rating by that of the study with the highest rating to arrive at a Q index $(Q_i)$ ranging from 0 to 1 (*Barendregt et al., 2013*). Studies of greater quality were given more weights, while heterogeneity was defined as $I2 > 75\%$. A funnel plot was used to determine the extent of the publishing bias.

## Additional research

Based on the existence of dental trauma, a sub-group analysis was carried out:

1) More than 3 mm or less than 3 mm of overjet

2) Lip coverage: is it enough or not enough?

3) Either an urban or rural residential area

4) School type: public or private

5) Children's body mass indices: obese or not.

Review Manager (RevMan) version 5.3 was used for the sub-group analysis.

## RESULTS

### Description of studies

After conducting the initial electronic database search on DOAJ and PubMed/MEDLINE, a total of 652 relevant titles related to the study issue were identified. There were 358 articles that were identified as duplicates. Following the abstract screening, 294 pertinent titles were chosen by two independent reviewers, and 230 were discarded since they had no bearing on the subject. The reviewers looked over and discussed 64 articles before choosing them for full-text analysis. Manual searches of the selected studies' reference lists turned up no further articles. Twenty-three studies were included in the data extraction and statistical analysis for the qualitative synthesis (Fig. 1).

### Components of included studies

All twenty-three studies used a descriptive cross-sectional design. These studies were conducted in different Arab countries with seven studies in Jordan conducted by *Hamdan & Rock (1995)*, *Al-Jundi (2002)*, *Ajlouni, Jaradat & Rihani (2010)*, *Rajab et al. (2013)*, *Al-Bajjali & Rajab (2014)*, *Al-Batayneh et al. (2017)*, and *Rajab & Abu Al Huda (2019)*, one in Syria led by *Marcenes et al. (1999)*, eight in Saudi Arabia conducted by *Al-Majed, Murray & Maguire (2001)*, *AlSarheed, Bedi & Hunt (2003)*, *Al-Malik (2009)*, *AlSadhan (2016)*, *Al-Ansari & Nazir (2020)*, *Shehri et al. (2021)*, *Basha et al. (2021)*, and *Alshammary et al. (2022)*, one in Kuwait led by *Artun et al. (2005)*, two in Palestine (occupied territories) by *Sgan-Cohen, Yassin & Livny (2008)*, *Muhamad, Nezar & Azzaldeen (2016)*, one in Iraq by *Noori & Al-Obaidi (2009)*, one in Egypt by *El-Kenany, Awad & Hegazy (2016)*, one in Libya by *Arheiam et al. (2020)* and one in UAE led by *Hashim et al. (2022)*. Overall, data on 35,390 children were included in this systematic review. Information about ethical approval for the study was not mentioned by seven studies (*Hamdan & Rock, 1995*; *Al-Jundi, 2002*; *Ajlouni, Jaradat & Rihani, 2010*; *Marcenes et al., 1999*; *Al-Majed, Murray & Maguire, 2001*; *Al-Malik, 2009*; *El-Kenany, Awad & Hegazy, 2016*). The most commonly used sampling technique in the primary studies was multi-stage stratified cluster sampling. Three of the included studies collected data retrospectively *i.e.*, from past records (*Al-Jundi, 2002*; *Ajlouni, Jaradat & Rihani, 2010*; *Shehri et al., 2021*). The remaining studies collected data using clinical examination of children and questionnaires (Table 1). Three of the included studies (*Al-Batayneh et al., 2017*; *AlSarheed, Bedi & Hunt, 2003*; *AlSadhan, 2016*) were comparative designs, in which the population of special children was compared with the control group. The disabilities considered were visually impaired, hearing impaired, and mentally disabled (Table 2).

The most common cause of oral injuries was a fall, followed by contact sports, physical altercations, and hard biting. The most common dental injuries caused by trauma were chips and cracks in the enamel, as well as cracks and chips in the dentin. Fractured teeth were an exceptionally uncommon cause of tooth loss. Playgrounds and schools were the most common places for serious dental injuries to occur. All research pointed to a maxillary impact, with special care given to the teeth in the maxillary anterior arch that had been damaged in the accident.

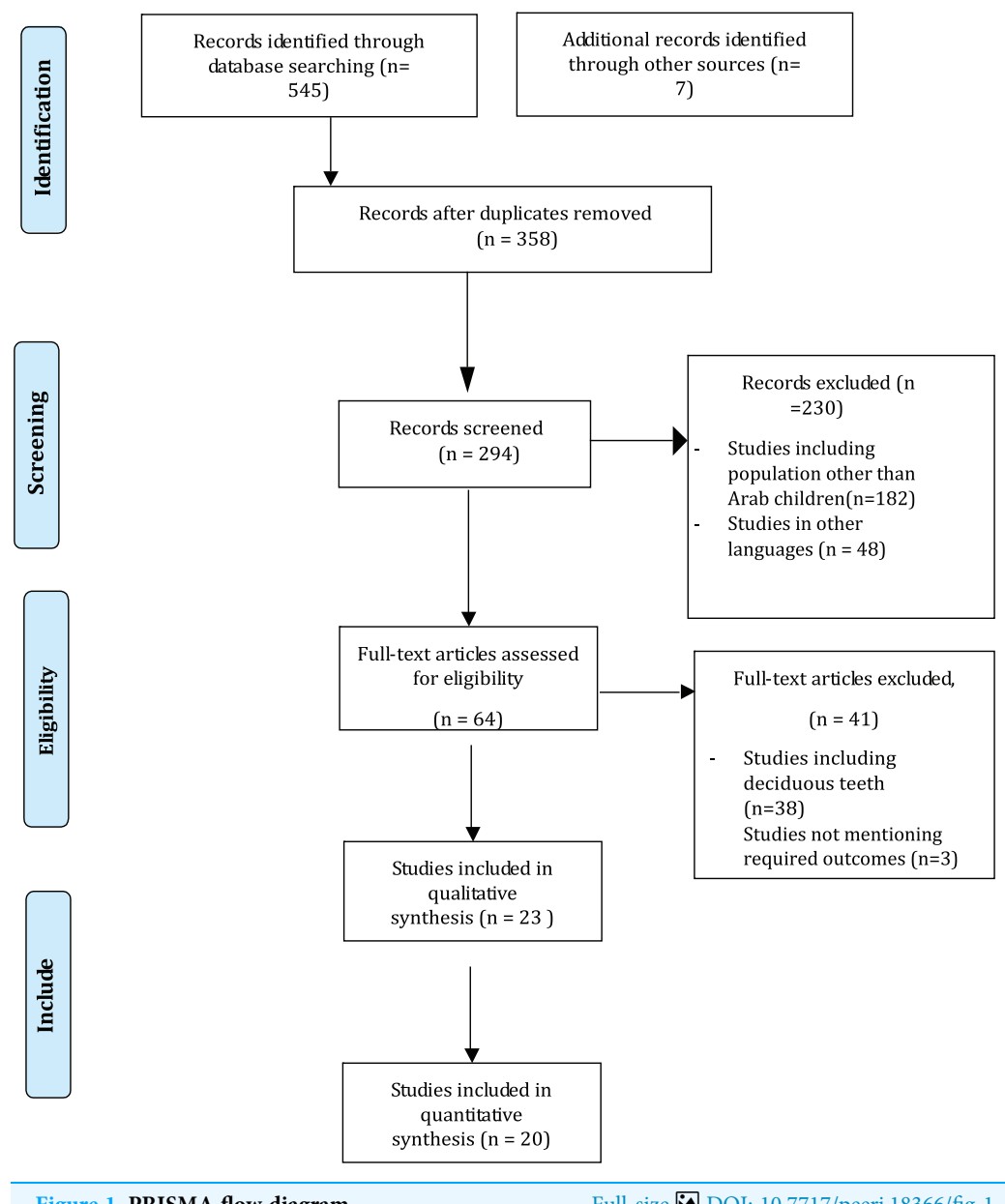

**Figure 1 PRISMA flow diagram.**

## Quality assessment of included studies

Among the 23 included studies, three studies (*Ajlouni, Jaradat & Rihani, 2010*; *Shehri et al., 2021*; *Muhamad, Nezar & Azzaldeen, 2016*) showed low quality while the remaining 20 studies showed high quality according to NOS. For studies with overall low quality, information about the representativeness of the sample, sample size, and validated screening tools for dental trauma was not mentioned as the data in these studies was collected retrospectively. The highest score of 5 was obtained by twelve studies that provided all the information particular to the domains of NOS (Table 3).

**Table 1 Characteristics of included studies.**

| Study ID | Place of study | Population | Age | Sample size | Gender (n) Male | Gender (n) Female | Prevalence | Cause of trauma % (n) | Type of trauma (%) | Conclusion |
|---|---|---|---|---|---|---|---|---|---|---|
| Hamdan & Rock (1995) | Amman, Jordan | School children | 10–12 years | 459 | 223 | 256 | 17.4% | Not mentioned | E only: 62.2 / ED: 23.5 | Enamel fractures were the most prevalent form of injury in both regions, with falls and playground accidents being the leading causes of these injuries. Injuries to the incisor teeth were more common among children whose overjet was larger than 5 mm than among those whose overjet was normal. |
| Marcenes et al. (1999) | Damascus, Syria | School children | 9–12 | 1,087 | 641 | 446 | 8% | Fall: 6.9 / Collision: 16 / Accidents: 24.1 / Violence: 42.5 / Others: 3.4 | E only – 5.6 / ED – 5.5 / EDP – 0.2 / Missing – 0 | Dentists have a responsibility to the public to advocate for legislation and educational campaigns to reduce the prevalence of traumatic dental injuries and to offer the knowledge policymakers need to establish safeguards to protect the public's oral health. |
| Al-Majed, Murray & Maguire (2001) | Riyadh | School children | 12–14 years | 862 | 862 | – | 34.3% | Not mentioned | E only fracture: 333/448 / ED fracture: 68/448 / EDP: 22/448 / Missing: 12/448 | Data of this study indicated that 12.6% of this study population had severe dental trauma. Gender, lip competency, and anterior incisal overjet were the primary predictors. No significant difference was found between Arab and Jewish youngsters when compared to a prior poll of the same kind. |
| Al-Jundi (2002) | Irbid, Jordan | Children attending dental emergency clinics | – | 620 | – | – | 31% | Fall – 80 / Sports – 3.1 / Collision – 7.7 / Fight – 7.7 / Accidents – 1.5 | E only: 2 / ED: 45.4 / EDP – 28.2 / Whole crown fracture – 1 | In order to improve the prognosis of traumatic injuries, it is essential to increase public dental knowledge of the devastating long-term consequences of such injuries. |
| Artun et al. (2005) | Kuwait | School children | 13–14 years | 1,583 | 788 | 795 | 14.5% | Not mentioned | E only: 193/288 / ED: 67/288 / Pulp damage: 0 / Crown repair: 18/288 / Pulp repair: 9/288 / Missing: 1/288 | Patients with OJ bigger than 9.5 mm, and those with OJ of 6.5 to 9.0 mm, had a 2.8-fold increased risk of maxillary incisor damage before puberty compared with subjects with normal OJ. |
| Sgan-Cohen, Yassin & Livny (2008) | Eastern Jerusalem | School children | 10–12 years | 452 | 272 | 181 | 33.8% | Fall – 29.1 / Activities – 16.4 / Cruelty – 20 / Gaming – 20 | Not mentioned | According to the results of this investigation, 12.6% of the population had experienced serious dental trauma. Gender, lip competency, and anterior incisal overjet were the primary predictors. |
| Al-Malik (2009) | Jeddah | Children reporting hospital | <17 years | 112 | 79 | 33 | 58.02% | Fall: 68.8 / Impact: 8.9 / Collision: 14.3 / Other: 8 | Not mentioned | This study's population revealed patterns and typical presentations of paediatric traumatic mouth injuries treated in a large Saudi Arabian hospital. Since this is an unavoidable health issue, we must increase public knowledge of oral trauma and highlight the significance of reducing its consequences via training and education. |
| Noori & Al-Obaidi (2009) | Sulaimani City, Iraq | School children | 6–13 years | 4,015 | 50.8% | 49.20% | 6.1% | Not mentioned | E only – 36.6 / ED – 35.4 / Concussion – 11.5 | While the number of children who had dental trauma was found to be relatively low in the current research, it still represents a sizable population. Furthermore, the number of children who required dental care after experiencing trauma was also shown to be rather significant. |

| Study ID | Place of study | Population | Age | Sample size | Male | Female | Prevalence | Cause of trauma % (n) | Type of trauma (%) | Conclusion |
|---|---|---|---|---|---|---|---|---|---|---|
| Ajlouni, Jaradat & Rihani (2010) | Jordan | Children referred to pediatric clinic | 6–14 years | 3,750 | – | – | 9.89% | Fall – 88.95<br>Accident – 0.54<br>Collision – 5.93<br>Sports – 4.04 | Not mentioned | Traumatic dental injury is most frequent between age 8 and 10 years. Males suffered dental injury more often than females. Falls accounted for most of injury causes in children. Majority of patients have had only one affected tooth. Crown fracture class-2 involving enamel, and dentine without pulp exposure is the most common crown fracture. The most injured tooth was maxillary central incisor. Right maxillary central incisor was affected more than the left side. |
| Rajab et al. (2013) | Amman, Jordan | School children | 12 years | 2,560 | 48% | 52% | 5.5% | Fall: 64.8 (92)<br>Sports: 19 (27)<br>Collision: 7.1 (11)<br>Violence 7 (10)<br>Traffic accident 1.4 (2) | E only: 37.3<br>ED fracture: 42.3 | There is a need for public interventions to reduce the risk for TDI among children in Jordan. |
| Al-Bajjali & Rajab (2014) | Amman, Jordan | School children | 12 years | 1,025 | 545 | 470 | 16% | Not mentioned | Not mentioned | The results of this epidemiological survey show that there is no significant association between TDIs and obesity among 12-year-old Jordanian school children, and inadequate lip coverage is the principal orofacial risk factor for TDIs |
| El-Kenany, Awad & Hegazy (2016) | Egypt | School children | 8–12 years | 7,983 | 50% | 50% | 14.6% | Falls: 38.3 (446)<br>Accident: 13.5 (157)<br>Sports: 10.9 (127)<br>Violence: 9.3 (108)<br>Home abuse: 0.9 (11)<br>Biting: 7.5 (88) | Not mentioned | There is a comparatively high incidence of missing permanent front teeth among students in grades 8–12 in the Dakahlia governorate of Egypt (14.6%). The most frequent form of TDIs was a single tooth fracture, usually involving the central incisors in the upper jaw. Falls were the leading cause of TDIs, and most of these accidents happened at school. |
| Muhamad, Nezar & Azzaldeen (2016) | Palestine (occupied territories) | School children | 9–12 years | 4,262 | 2,344 | 1,918 | 12.2% | Not mentioned | Not mentioned | Overall traumatised permanent incisors were found to occur fairly frequently with males having experienced significantly more TDIs than females. The prevalence of TDIs was 12.2%. Falls was the most common reason. |
| Rajab & Abu Al Huda (2019) | Amman, Jordan | School children | 12 years | 1,652 | 796 | 856 | 14.6% | Not mentioned | E: 69.8 (169)<br>ED: 14.5 (35)<br>Pulp: 2.1 (5)<br>Missing: 0 | The prevalence of TDI was 14.6%. Untreated TDI had a negative impact on OHRQoL compared to absence of or treated TDI among 12-year-old school children in Amman. |
| Al-Ansari & Nazir (2020) | Al-Khobar, Dammam, and Dhahran | School children | 12–15 years<br>Mean: 14.29 +–1.11 | 258 | 258 | – | 39.5% | Fall on the ground 9.3%<br>Accidental hits 8.9%<br>Violence 4.2% | Not mentioned | This study indicated that dental trauma was very common among school children. However, few with dental trauma received dental treatment. Immediate care of dental trauma was uncommon among children |
| Arheiam et al. (2020) | Benghazi, Libya | School children | 12 years | 1,134 | 552 | 582 | 10.3% | Not mentioned | Not mentioned | The study revealed that 10.3% of Libyan children aged 12 had TDIs, with substantial unmet treatment requirements. Improved dental care for children with TDIs in Libya requires further work to create effective preventative programmes. |

(Continued)

| Study ID | Place of study | Population | Age | Sample size | Gender (n) Male | Gender (n) Female | Prevalence | Cause of trauma % (n) | Type of trauma (%) | Conclusion |
|---|---|---|---|---|---|---|---|---|---|---|
| *Shehri et al.* (2021) | Riyadh | Pediatric emergency unit | <12 years | 223 | 132 | 91 | 100% | Accident: 8.2% Violence: 1.4% Fall: 64.4% Sports 1.4% Unknown: 24.7% | Not mentioned | The research also highlighted the need of adding specialised personnel to the OMFS team to better treat juvenile patients, such as a pedodontist and a general dentist. |
| *Basha et al.* (2021) | Taif | Special needs children | 6–16 | 350 | 131 | 219 | 25.4% | Not mentioned | E only: 51.3% ED without pulp: 30.2% Crown fracture: 10.1% Crown fracture with pulp exposure: 1.7% Missing due to trauma: 0 E + ED fracture: 4.2% | Increased overjet, insufficient lip covering, and cerebral palsy were shown to have a strong correlation with TDIs in the current investigation. |
| *Alshammary et al.* (2022) | Ha'il | Children | 6–15 years | 555 | – | – | 44% | Fall: 61.15% Violence: 7.44% Sports: 9.11% Accident: 4.54% Biting hard objects: 11.15% Others: 6.61% | Not mentioned | The significance of oral trauma and its related risk factors in the setting of Ha'il, Saudi Arabia, has been shown by this research. The results of this survey show that over half of the respondents report their children had experienced oral trauma. Most oral trauma is the result of a fall, and homes are the most prevalent sites of injury. |
| *Hashim et al.* (2022) | Ajman | School children | 12 years | 1,008 | 510 | 498 | 9.8% | Fall: 42.4% Collision: 53.5% Sport: 1% Other causes: 3% | E: 58.7% ED: 34.3% ED without pulp: 3% Non-vital: 0 Displacement: 1% Tooth loss: 1% Fracture and restoration: 2% | The findings illustrated that the prevalence of traumatic dental injury to the permanent incisors of 12-year-old children was low. Being male, having a mother with a low level of education, having an overjet more than 5 mm, and having an inadequate lip coverage were significantly related to the prevalence of TDIs among children in the United Arab Emirates |

**Note:**

E, enamel only; ED, enamel and dentin; EDP, enamel, dentin, pulp.

**Table 2 Characteristics of studies with a comparison group.**

| Study ID | Place of study | Population | Age | Sample size | Criteria used for trauma assessment | Prevalence Case | Control | Type of trauma (%) | Conclusion |
|---|---|---|---|---|---|---|---|---|---|
| AlSarheed, Bedi & Hunt (2003) | Riyadh | Visual impaired | 11–16 years | 77 | Clinical examination trauma index according to BASCD | Visual impaired 9% | 6.70% | E only: 3.4 | Kids with sensory impairments are more likely to have oral trauma. However, this was much higher for HI kids solely. Children in the VI group did not show the same gender difference in exposure to trauma as those in the HI group, where it was more pronounced among males. |
| | | Hearing impaired | | 210 | | | | E and D: 1.8 | |
| | | Controls | | 494 | | Hearing impaired 11.4% | | | |
| AlSadhan (2016) | Riyadh | Institutionalized orphans | 4–12 years | 90 | Not mentioned | 19/90 | 9/90 | – | The findings indicated that orphans had an increased prevalence of oral habits and dental trauma, suggesting underlining psychological disorders. The effect of oral habits on the occlusal status of orphan children influenced the facial appearance and could result in low self-esteem. |
| | | Controls | | 90 | | | | | |
| Al-Batayneh et al. (2017) | Jordan | Special school children | 11.8 + −4.2 years | 959 | Questionnaire and clinical examination | 8.7% | 4.1% | Cases: E – 53.2%, ED – 32%, EDP – 9% Controls: E – 55.6%, ED – 33.3%, EDP – 9.7% | A higher prevalence of TDI was found in CSHCN especially in children with multiple disabilities, followed by intellectual disabilities, and children with cerebral palsy. |
| | | Controls | | 1,010 | | | | | |

**Table 3 The Newcastle–Ottawa scale is modified for use in cross-sectional investigations.**

| Study ID | Representativeness of sample | Sample size | Comparability | Validated screening tools | Adequacy of descriptive statistics | Total score | Quality |
|---|---|---|---|---|---|---|---|
| Hamdan & Rock (1995) | * | * | * | * | * | 5 | High |
| Marcenes et al. (1999) | * | * | * | * | * | 5 | High |
| Al-Majed, Murray & Maguire (2001) | * | * | * | * | * | 5 | High |
| Al-Jundi (2002) | – | * | * | * | * | 4 | High |
| AlSarheed, Bedi & Hunt (2003) | – | – | * | * | * | 3 | High |
| Artun et al. (2005) | * | * | * | * | * | 5 | High |
| Sgan-Cohen, Yassin & Livny (2008) | * | * | * | * | * | 5 | High |

(Continued)
| Table 3 (continued) | | | | | | | |
|---|---|---|---|---|---|---|---|
| Study ID | Representativeness of sample | Sample size | Comparability | Validated screening tools | Adequacy of descriptive statistics | Total score | Quality |
| Al-Malik (2009) | – | – | * | * | * | 3 | High |
| Noori & Al-Obaidi (2009) | – | * | * | * | * | 4 | High |
| Ajlouni, Jaradat & Rihani (2010) | – | – | * | – | * | 2 | Low |
| Rajab et al. (2013) | * | * | * | * | * | 5 | High |
| Al-Bajjali & Rajab (2014) | * | * | * | * | * | 5 | High |
| AlSadhan (2016) | – | – | * | * | * | 3 | High |
| El-Kenany, Awad & Hegazy (2016) | * | * | * | * | * | 5 | High |
| Muhamad, Nezar & Azzaldeen (2016) | – | – | – | * | * | 2 | Low |
| Al-Batayneh et al. (2017) | – | – | * | * | * | 3 | High |
| Rajab & Abu Al Huda (2019) | * | * | * | * | * | 5 | High |
| Al-Ansari & Nazir (2020) | * | * | * | * | * | 5 | High |
| Arheiam et al. (2020) | * | * | * | * | * | 5 | High |
| Shehri et al. (2021) | – | – | * | – | * | 2 | Low |
| Basha et al. (2021) | * | * | * | * | * | 5 | High |
| Alshammary et al. (2022) | * | – | * | * | * | 4 | High |
| Hashim et al. (2022) | * | * | * | * | * | 5 | High |

## META-ANALYSIS

### Prevalence of dental trauma

Twenty-two studies found that dental trauma was common. The frequency of dental trauma in Arab children was calculated to be 26%, with a 95% confidence interval of 10% to 43%. Results were pooled using a random effects model due to high levels of heterogeneity ($I^2$ = 99.9%) and significance level ($p < 0.05$) as presented in Fig. 2.

### Damage to the teeth and overjet

In ten experiments, researchers looked at how oral trauma affected overjet. Children with overjet more than 3.5 mm were shown to be 1.78 times more likely to have dental injury than children with normal overjet (pooled odds ratio 1.78; 95% confidence range [1.17–2.70]). The data was 93% heterogeneous and significantly different from one another ($p < 0.05$), hence a random effects model was used (Fig. 3).

### Lip coverage and dental trauma

Eight studies looked at the link between lip protection and tooth damage. Children with inadequate lip covering had an odds ratio of 2.57 times more likely to have oral trauma than children with adequate lip coverage (95% CI [1.82–3.62]). That was considerable ($p <$

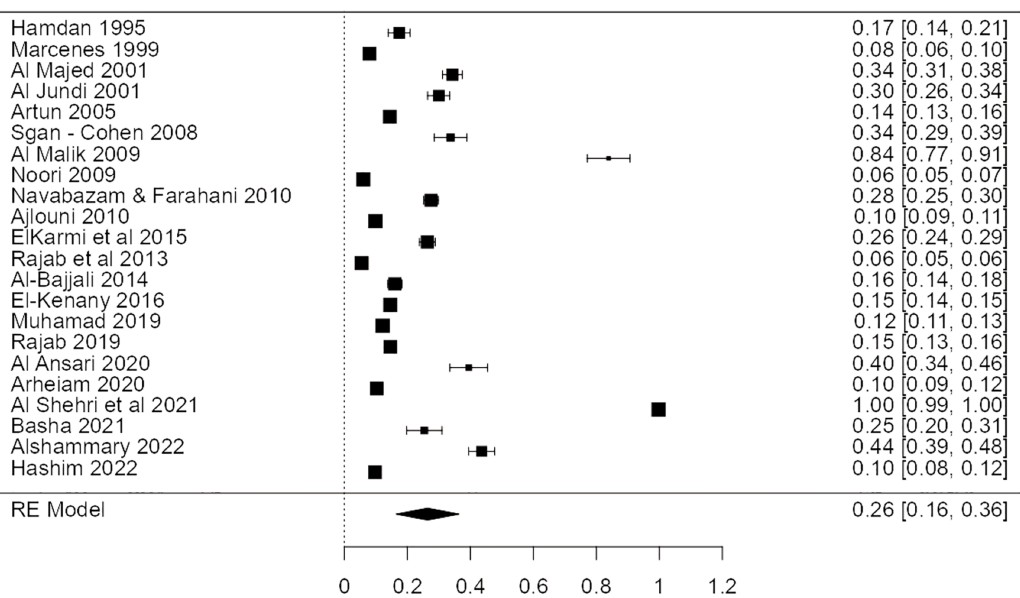

| | | | | | | | |
|---|---|---|---|---|---|---|---|
| Hamdan 1995 | | | | | | | 0.17 [0.14, 0.21] |
| Marcenes 1999 | | | | | | | 0.08 [0.06, 0.10] |
| Al Majed 2001 | | | | | | | 0.34 [0.31, 0.38] |
| Al Jundi 2001 | | | | | | | 0.30 [0.26, 0.34] |
| Artun 2005 | | | | | | | 0.14 [0.13, 0.16] |
| Sgan - Cohen 2008 | | | | | | | 0.34 [0.29, 0.39] |
| Al Malik 2009 | | | | | | | 0.84 [0.77, 0.91] |
| Noori 2009 | | | | | | | 0.06 [0.05, 0.07] |
| Navabazam & Farahani 2010 | | | | | | | 0.28 [0.25, 0.30] |
| Ajlouni 2010 | | | | | | | 0.10 [0.09, 0.11] |
| ElKarmi et al 2015 | | | | | | | 0.26 [0.24, 0.29] |
| Rajab et al 2013 | | | | | | | 0.06 [0.05, 0.06] |
| Al-Bajjali 2014 | | | | | | | 0.16 [0.14, 0.18] |
| El-Kenany 2016 | | | | | | | 0.15 [0.14, 0.15] |
| Muhamad 2019 | | | | | | | 0.12 [0.11, 0.13] |
| Rajab 2019 | | | | | | | 0.15 [0.13, 0.16] |
| Al Ansari 2020 | | | | | | | 0.40 [0.34, 0.46] |
| Arheiam 2020 | | | | | | | 0.10 [0.09, 0.12] |
| Al Shehri et al 2021 | | | | | | | 1.00 [0.99, 1.00] |
| Basha 2021 | | | | | | | 0.25 [0.20, 0.31] |
| Alshammary 2022 | | | | | | | 0.44 [0.39, 0.48] |
| Hashim 2022 | | | | | | | 0.10 [0.08, 0.12] |
| RE Model | | | | | | | 0.26 [0.16, 0.36] |

**Figure 2** Forest plot demonstrating the incidence of dental trauma in Arab children's permanent teeth (*Hamdan & Rock, 1995*; *Marcenes et al., 1999*; *Al-Majed, Murray & Maguire, 2001*; *Al-Jundi, 2002*; *Artun et al., 2005*; *Sgan-Cohen, Yassin & Livny, 2008*; *Al-Malik, 2009*; *Noori & Al-Obaidi, 2009*; *Navabazam & Farahani, 2010*; *Ajlouni, Jaradat & Rihani, 2010*; *ElKarmi et al., 2015*; *El-Kenany, Awad & Hegazy, 2016*; *Muhamad, Nezar & Azzaldeen, 2016*; *Rajab & Abu Al Huda, 2019*; *Al-Ansari & Nazir, 2020*; *Arheiam et al., 2020*; *Shehri et al., 2021*; *Basha et al., 2021*; *Alshammary et al., 2022*; *Hashim et al., 2022*).

| Study or Subgroup | More than 3.5mm Events | Total | Normal Events | Total | Weight | Odds Ratio M-H, Random, 95% CI | Odds Ratio M-H, Random, 95% CI |
|---|---|---|---|---|---|---|---|
| Al-Bajjali 2014 | 80 | 417 | 85 | 598 | 10.6% | 1.43 [1.03, 2.00] | |
| Al Majed 2001 | 122 | 302 | 174 | 560 | 10.8% | 1.50 [1.12, 2.01] | |
| Arheiam 2020 | 61 | 443 | 56 | 691 | 10.4% | 1.81 [1.23, 2.66] | |
| Artun 2005 | 92 | 532 | 135 | 979 | 10.8% | 1.31 [0.98, 1.75] | |
| Basha 2021 | 49 | 93 | 32 | 257 | 9.5% | 7.83 [4.52, 13.58] | |
| El-Kenany 2016 | 310 | 1346 | 856 | 6637 | 11.3% | 2.02 [1.75, 2.34] | |
| Hamdan 1995 | 19 | 80 | 61 | 80 | 8.5% | 0.10 [0.05, 0.20] | |
| Hashim 2022 | 82 | 462 | 17 | 546 | 9.6% | 6.71 [3.92, 11.51] | |
| Marcenes 1999 | 10 | 69 | 77 | 1018 | 8.6% | 2.07 [1.02, 4.21] | |
| Sgan Cohen 2008 | 39 | 79 | 114 | 374 | 9.9% | 2.22 [1.36, 3.64] | |
| **Total (95% CI)** | | **3823** | | **11740** | **100.0%** | **1.78 [1.17, 2.70]** | |
| Total events | 864 | | 1607 | | | | |

Heterogeneity: Tau² = 0.40; Chi² = 123.22, df = 9 (P < 0.00001); I² = 93%
Test for overall effect: Z = 2.70 (P = 0.007)

More than 3.5mm   Normal

**Figure 3** Forest plot demonstrating a connection between tooth trauma and overjet (*Al-Bajjali & Rajab, 2014*; *Al-Majed, Murray & Maguire, 2001*; *Arheiam et al., 2020*; *Artun et al., 2005*; *Basha et al., 2021*; *El-Kenany, Awad & Hegazy, 2016*; *Hamdan & Rock, 1995*; *Hashim et al., 2022*; *Marcenes et al., 1999*; *Sgan-Cohen, Yassin & Livny, 2008*).

| Study or Subgroup | Inadequate Events | Total | Adequate Events | Total | Weight | Odds Ratio M-H, Random, 95% CI |
|---|---|---|---|---|---|---|
| Al-Bajjali 2014 | 114 | 551 | 51 | 464 | 13.9% | 2.11 [1.48, 3.02] |
| Arheiam 2020 | 24 | 200 | 93 | 934 | 12.5% | 1.23 [0.77, 1.99] |
| Artun 2005 | 7 | 61 | 61 | 424 | 8.4% | 0.77 [0.34, 1.77] |
| Basha 2021 | 51 | 105 | 30 | 245 | 11.7% | 6.77 [3.94, 11.62] |
| El-Kenany 2016 | 672 | 2849 | 494 | 5134 | 16.0% | 2.90 [2.55, 3.29] |
| Hashim 2022 | 77 | 417 | 22 | 591 | 12.3% | 5.86 [3.58, 9.58] |
| Marcenes 1999 | 29 | 194 | 58 | 893 | 12.5% | 2.53 [1.57, 4.07] |
| Sgan Cohen 2008 | 50 | 102 | 103 | 351 | 12.8% | 2.32 [1.47, 3.63] |
| **Total (95% CI)** | | **4479** | | **9036** | **100.0%** | **2.57 [1.82, 3.62]** |
| Total events | 1024 | | 912 | | | |

Heterogeneity: Tau² = 0.19; Chi² = 42.93, df = 7 (P < 0.00001); I² = 84%
Test for overall effect: Z = 5.35 (P < 0.00001)

**Figure 4** A forest plot demonstrating the relationship between lip protection and dental trauma (*Sgan-Cohen, Yassin & Livny, 2008*; *Marcenes et al., 1999*; *Hashim et al., 2022*; *El-Kenany, Awad & Hegazy, 2016*; *Basha et al., 2021*; *Artun et al., 2005*; *Arheiam et al., 2020*; *Al-Bajjali & Rajab, 2014*).

0.05) variability in the data (84%), necessitating the employment of a random effects model (Fig. 4).

## Oral injury and gender

Fifteen studies looked at how different sexes were affected by oral trauma. The combined odds ratio found was 2.06 (95% confidence interval [CI][1.72 to 2.47]), meaning that boys used to have a 2.06-fold greater risk of dental trauma than girls. There was high heterogeneity in the data (76%), hence a random effects model was used to analyse the data ($p < 0.05$) as presented in Fig. 5.

## Dental injuries and academic level

There were three studies in total. The pooled odds ratio found was 4.17 (95% CI [0.54–32.43]), meaning that public school students had 4.17 times the odds of dental trauma as compared to private school students. The results were 99% heterogeneous and significant enough to justify a random effects model ($p < 0.05$) presented in Fig. 6.

## Dental pain and a residential setting

There were three research that analysed how living situation is related to dental injury. Children in rural regions had a combined odds ratio of 0.82 (95% CI [0.24–0.63]), indicating an 8.82-fold reduced incidence of dental trauma compared to children in urban areas. There was high heterogeneity in the data (87%), thus a quantitative technique was used and the findings were considerable ($p < 0.05$) (Fig. 7).

## Tooth damage and BMI

Three studies analysed the correlation between oral trauma and body mass index. Children who were overweight were less likely to have dental damage than children who were not overweight, as shown by a combined odds ratio of 0.71 (95% CI [0.28–1.84]). These results

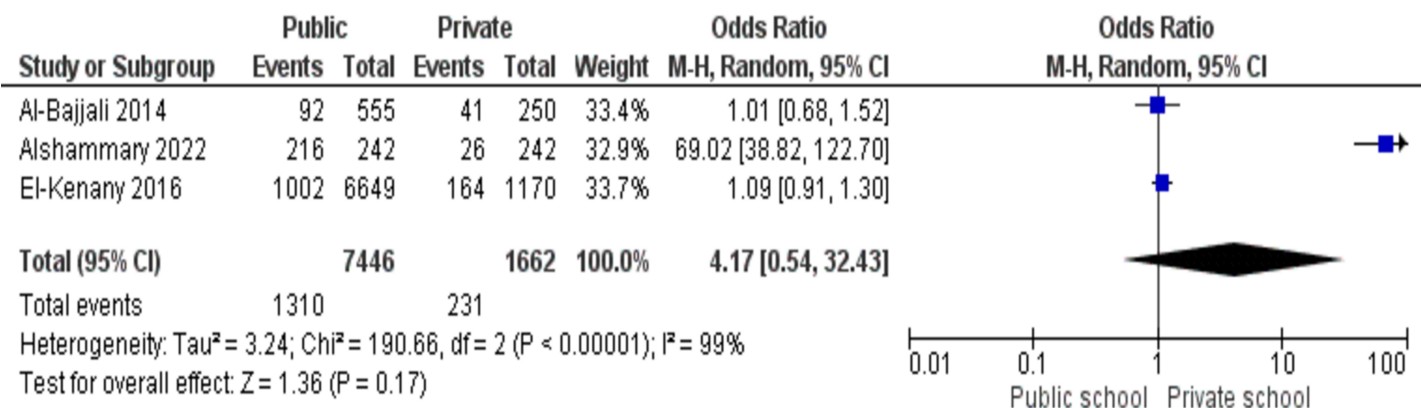

**Figure 5** Forest plot indicating the relationship between gender and dental trauma (*Ajlouni, Jaradat & Rihani, 2010*; *Al-Bajjali & Rajab, 2014*; *Al-Malik, 2009*; *Alshammary et al., 2022*; *Shehri et al., 2021*; *Arheiam et al., 2020*; *Artun et al., 2005*; *Basha et al., 2021*; *Hashim et al., 2022*; *Marcenes et al., 1999*; *Muhamad, Nezar & Azzaldeen, 2016*; *Navabazam & Farahani, 2010*; *Noori & Al-Obaidi, 2009*; *Rajab & Abu Al Huda, 2019*; *Sgan-Cohen, Yassin & Livny, 2008*).

**Figure 6** Association between the type of school and dental trauma shown by a forest plot (*El-Kenany, Awad & Hegazy, 2016*; *Alshammary et al., 2022*; *Al-Bajjali & Rajab, 2014*).

are statistically significant ($p < 0.05$), and the 83% heterogeneity necessitated the employment of a random effects model (Fig. 8).

## Publication bias

Each included study is represented by the dot. The Y-axis represents standard error representing study precision where as X-axis shows the estimated effect size, in this case proportions. As shown in Fig. 9, the present funnel plot is asymmetrical. Majority of the

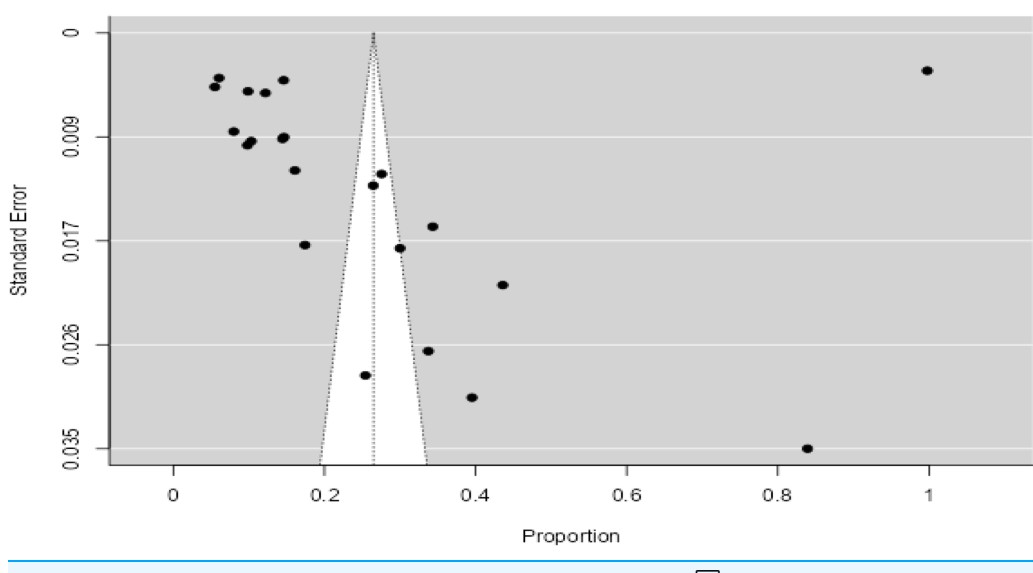

**Figure 7** Forest plot indicating a relationship between residential location and dental trauma (*Al-Bajjali & Rajab, 2014*; *El-Kenany, Awad & Hegazy, 2016*; *Rajab & Abu Al Huda, 2019*).               

**Figure 8** Forest plot showing the association between BMI and dental trauma (*Al-Bajjali & Rajab, 2014*; *Arheiam et al., 2020*; *Basha et al., 2021*).               

**Figure 9** Publication bias.               

studies are seen outside the funnel indicating publication bias. There are less than 95% studies present inside the diagonal dotted 95% confidence interval line.

## DISCUSSION

Care should be taken when comparing the prevalence levels of TDI because different researchers have used different sample procedures, dentition types, and diagnostic standards. The current study focused on 12-year-old children to allow comparison with other studies from throughout the world that have been published. This research has several restrictions. For instance, only injuries linked to the teeth were reported because soft tissue injuries during the clinical examination could not be noted. Furthermore, as the study's results were self-reported, it's possible that the causes of TDIs were not correctly reflected. Injury episodes are not purely arbitrary; they frequently have predictable causes. Boys were more likely than girls to sustain traumatic dental injuries, which is in line with other researchers' findings (*Al-Bajjali & Rajab, 2014*; *Telford et al., 2016*). Due to their higher degree of behaviour maturity than males, who tend to engage in risk-taking behaviours, girls exhibit a lower occurrence of tooth trauma, which may help to explain why. The control that traditional families exercise over the girls' behaviour as a result of Arabs' cultural and religious attitudes is another possible explanation for the low prevalence of TDIs among girls in Arab countries (*Hashim, Williams & Murray Thomson, 2009*). Few studies on dental trauma take into account socioeconomic considerations. According to the results of Borges and co-workers (*Borges et al., 2017*), only the mothers' education levels in this study showed a significant correlation with the occurrence of severe dental injuries. The association between socioeconomic position and the likelihood of experiencing trauma is not well understood in the literature, and the research that has looked into it has come to different conclusions (*Oldin et al., 2016*; *Magno et al., 2020*). The fact that the earlier research was carried out in several nations with diverse socioeconomic status patterns may be partly to blame for these contradictory results. To better understand these linkages, more study is required. Increased incisor overjet was found to be a significant predisposing factor for traumatic dental damage, according to the current study's findings, which are in line with those of recently published systematic reviews (*Tewari et al., 2020*).

*Arraj, Rossi-Fedele & Doğramacı (2019)* observed that there was an association between overjet and dental trauma through a meta-analysis, They reported an odds of 3.37 in children 0–6 years of age with an overjet of 3 mm or greater. Children in mixed and secondary dentition with an overjet over 5 mm had an odds ratio of 2.43.

Inadequate lip coverage among schoolchildren greatly increased their chances of developing TDI, proving that it is a risk factor for TDI (*Abdel Malak et al., 2021*). This study demonstrates unequivocally that collisions and falls were the primary causes of traumatic oral injury. These results concur with those of earlier articles (*Dharmani, Pathak & Sidhu, 2019*; *Al-Jundi, 2002*). It is crucial to keep in mind that an "accident" could result in a tiny act of violence. Collisions can also happen during play or a sporting event; in any instance, they might be categorized as falls or sports injuries. Establishing a precise international standard for recording the aetiology of dental injuries is essential for enabling

accurate comparisons between countries. In developing solutions to safeguard oral health, it should always be taken into account that the conduct of the children may be directly tied to the appearance of the TDIs. A TDI is frequently connected with behavioural risk factors including violent attitudes (*Adekoya-Sofowora et al., 2009*). These were the primary activities that took place at home and at school where the majority of the traumatic dental injuries occurred; sports like swimming, football, or running were the main activities that were directly linked to the collisions and falls that resulted in the majority of the children's traumatic dental injuries. Other researchers (*Damé-Teixeira et al., 2013*) reported experiencing similar outcomes. As a result, social education at home and in schools should receive special attention. Less traumatic dental injuries will occur in schools when there is a supportive social and physical environment (*Eltair et al., 2020*). Activities aimed at prevention should be directed toward students. Our findings indicate that it is critical to take into account the methods for lowering the incidence of severe oral injuries. Mouth guard use should be advocated among kids who play contact sports. In order to reduce the likelihood of falls resulting in serious dental injuries in children, it is also imperative that they be closely watched during physical activity. Raising public awareness is necessary for controlling and reducing the prevalence of TDIs in children in the Arab region; this can be accomplished by implementing national and local activities. The majority of TDIs in the included sample simply produced enamel damage and did not need to be repaired. For dental specialists, restoring a tooth that has been damaged is a daily task (*Dharmani, Pathak & Sidhu, 2019*). Paediatric restorative dentistry requires a lot of time since children have variable levels of compliance and are frequently distracted. Despite the recent introduction of new "bioactive" materials that address these issues, it's critical to remember that the success of the restorative treatment should not be connected only to the type of material that is chosen, but also to the modification in the child's and the family's lifestyle (*Lardani et al., 2022*). However, almost one-third of TDIs caused dentine injury, and 3% of them had an impact on the enamel, dentine, and pulp, in agreement with *Firmino et al. (2014)* and *de Paiva et al. (2015)*. This demonstrated the substantial unmet need for dental care. Therefore, in order to promote observable improvements in oral health and to reduce the number of inequalities, in addition to preventive efforts that emphasize the causes of oral problems, appropriate post-traumatic dental therapy should be taken into account. Strategies that attempt to enhance the school setting and raise public awareness, particularly of educators who participate in athletic activities, may help to decrease the frequency of TDIs. Understanding the aetiology of traumatic injuries and their early prevention may lead to a decrease in the incidence of traumatic injuries. For example, early diagnosis and appropriate treatments undertaken by Clinicians may assist in limiting traumatic dental injuries. Raising awareness among public health authorities regarding the substantial influence of TDI management on the broader population is crucial; however, disease prevention—specifically through interceptive orthodontic treatment, the promotion of mouthguard usage, or a combination of both—primarily within the primary dental care context, could serve as a foundational strategy for diminishing TDI incidence within a framework of optimized resources aimed at addressing disparities in oral health status and inequities in access to oral health care (*Abbott, 2018*).

Traumatic oral injuries may have a significant impact on a child's quality of life; thus, they should not be disregarded (*Freire-Maia et al., 2015*; *Gherunpong, Tsakos & Sheiham, 2004*). Further study is needed to examine the social and personal factors that enhance the TDI risk in order to create and implement a successful preventative strategy to decrease the impact of TDIs in the future.

## CONCLUSION

Significant correlations between TDI and the male gender, increased overjet, and inadequate lip coverage were discovered. A qualitative synthesis also showed a positive correlation between TDI and sociodemographic elements like poor socioeconomic position and rural population, as well as environmental elements like TDI source and location. The data could not be combined in a way that satisfied the requirements for a meta-analysis, though. The prevalence of TDI in children and adolescents in the Arab region appears to be rising over time. Future population-based analytical research should concentrate on documenting the incidence and/or prevalence of TDI among marginalised communities in order to better understand the primary causes of TDI.

### Funding
Ajman University provided funding for this study in covering the article processing charges. The funders had no role in study design, data collection and analysis, decision to publish, or preparation of the manuscript.

### Grant Disclosures
The following grant information was disclosed by the authors:
Ajman University.

### Competing Interests
The authors declare that they have no competing interests.

### Author Contributions
- Raghad Hashim conceived and designed the experiments, performed the experiments, analyzed the data, authored or reviewed drafts of the article, and approved the final draft.
- Alexander Maniangat Luke conceived and designed the experiments, performed the experiments, authored or reviewed drafts of the article, and approved the final draft.
- Afraa Salah conceived and designed the experiments, analyzed the data, prepared figures and/or tables, and approved the final draft.
- Simy Mathew conceived and designed the experiments, prepared figures and/or tables, and approved the final draft.

### Data Availability
This is a systematic review and meta-analysis.

## Supplemental Information

Supplemental information for this article can be found online at http://dx.doi.org/10.7717/peerj.18366#supplemental-information.

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
