# Peer review of "Traumatic dental injuries in permanent teeth among Arab children: prevalence, and associated risk factors—a systematic review and meta-analysis"

_PeerJ, doi:10.7717/peerj.18366_

## Round 0.1 · original submission · Minor Revisions

Dear authors,
Please address the corrections indicated by our external peer reviewers. Additionally, I recommend to merge the sub-section titled "Research Rationale" within the introduction section. Ensure that you respond thoroughly to all questions and comments to avoid further rounds of review.

Reviewer 1 ·

Basic reporting

Firstly, I would like to thank the authors for submitting the manuscript to PeerJ. Although I find it very interesting, please find some suggestions and changes to further enhance its' quality:
1. Verify the abstract's length and ensure it is written as a unified whole. Strive to be concise while making sure all important aspects of the research are included.
2. Prior to introducing an abbreviation in the document for the first time, provide a full explanation and consistently use it throughout the text.
3. Verify if the citations are accurately referenced in the entire text and in the References section. Make sure that all citations are structured in accordance with the instructions of the journal.
4. Make sure that all abbreviations utilized in Tables 1 and 2 are clarified under the tables to assist readers' comprehension.
5. Elaborate on Figure 9 found in the passage and include a thorough description in the accompanying caption. This will assist readers in comprehending the importance and background of the figure.
6. Conduct a comprehensive review of grammar and spelling.

By covering these aspects, the manuscript will be greatly enhanced, becoming a stronger and more valuable addition to the field of dental injury research.

Kind regards!

Experimental design

It is advisable to include the manufacturer, city, and country information for the statistical programs and methods used, if available. This guarantees clarity and enables the analysis to be replicated.

Validity of the findings

Include citations for the strategy and techniques employed in the Research Rationale and Materials and Methods sections. This will improve the trustworthiness and replicability of the research.

Additional comments

The manuscript: "Prevalence, trends, and risk factors associated with traumatic dental injuries in permanent teeth among Arab children – a systematic review and meta-analysis" thoroughly examines traumatic dental injuries (TDIs) in Arab children by conducting a systematic review and meta-analysis, discussing prevalence, patterns, and related risk factors. This thorough strategy provides important understanding in a neglected field. Utilizing such a systematic review methodology guaranteed a comprehensive study that reduced bias, ultimately improving the reliability of the results. Effective use of tables and figures for data presentation aided in visualizing trends and associations, providing clear data representation.

Reviewer 2 ·

Basic reporting

This article is significant as it highlights the prevalence of such injuries among children and identifies the factors influencing their occurrence. The insights provided are crucial for enhancing treatment approaches and developing preventive strategies. This information is particularly relevant for shaping policies and guidelines related to dental trauma. Generally, the article is written in good and technically correct English, and literature references and article structure are valid and coherent within the scope of the review. However, I have reviewed the article and proposed the recommendations and suggestions as listed.

Experimental design

Abstract

• It is recommended to include the current research gap as the purpose of the study to better relate to the title and separate the methods from AIM and OBJECTIVE.

• Line 8 Result; It is advisable to perform a descriptive analysis of the total publications considered in the study and the types of studies included. Summarize each domain's application from the articles, including the primary and secondary outcomes of the study.

• Line 15-17 “For the studies that were considered, trend analysis was done. Utilizing the Hoy criteria for prevalence studies, the listed studies’ quality was evaluated. When meta-analysis could not be done for a predictor, a qualitative synthesis was conducted.”- It is advisable to clearly state the outcome measures or data collected that reflect the results section.


• Line 23-25 Conclusion; “The prevalence of TDI in children and adolescents in the Arab region appears to be rising over time. Future population-based analytical research should concentrate on documenting the incidence and/or prevalence of TDI among marginalised communities in order to better understand the primary causes of TDI.”- This conclusion is too general and does not adequately reflect the results in the abstract.


Methods

• The methodology of the review is clearly stated. Wide description has been given to the search strategy, as well as to the most used systems for systematic reviews. Assessment of risk of bias appropriate. However..,
• Search strategy/ Searching databases/ Screening/ Research question- Please explain why the Scopus database was not used, as it contains numerous articles related to dentistry.

• Line 101-102- ‘ consisted of kids from Arab nations’ It is advisable to state the age limits included in this study or any exclusion criteria for the population/subjects.

• Line 102-103: “the outcome was the frequency with which such injuries occurred, and the trials were cross-sectional in nature.” The importance of the outcome is to be tabulated and data extracted presented and declared. To provide more explanation about the outcome of your study (Main outcomes and Secondary outcome) (specifically, you may expand upon the data collected, frequency, percentages, and mean value)

• Forming Criteria-Do this article have exclusion criteria of outcomes?
o Advisable to give inclusion and exclusion for article criteria to be included.
o Inclusion and exclusion criteria advisable to be included for
 Type of Study
 Eligibility Criteria and Content


• Protocol registration - The review protocol was registered with PROSPERO (ID number CRD42023421734).- 7/05/2023
It is advisable to declare the period of publications considered in this article or the date of the last publication update.

• To declare if there was a dispute between reviewers, what is the solution for the article to be rejected or included?

• Quality assessment, Risk of bias in individual studies/ Risk of bias assessment/ the Newcastle-Ottawa scale modified- No reference or citation.
• Data analysis. ‘The random-effect model’ has no citation or reference
o Line 157-159 ; “The QE model was used to synthesize the pooled prevalence based on the quality score given to each research. The quality of each study was quantified by dividing its quality rating by that of the study with the highest rating to arrive at Q index (Qi) ranging from 0 to 1.” The reference source for this system was not mentioned. It is advisable to place the reference.

Validity of the findings

Result & Discussion
• Line 203-209- Where are the result references to which table and the finding not refine?

Conclusion
• Line 340: “The majority of research showed a low to high risk of bias”. It is advisable to remove this statement from the conclusion as it pertains more to the descriptive results of the risk of bias.

Table/figure
• Figure 2: Forest plot demonstrating the incidence of dental damage in Arab children's permanent teeth- It is advisable to standardize the term as either "dental trauma" or "dental injury."

Additional comments

Dear Editorial Team and Authors,

The original study titled "Traumatic Dental Injuries in Permanent Teeth Among Arab Children: Prevalence, Trends, and Associated Risk Factors – A Systematic Review and Meta-Analysis" presents a comprehensive review of cross-sectional studies conducted in Arab countries, focusing on the prevalence of traumatic dental injuries (TDI) in permanent teeth.

This study is significant as it highlights the prevalence of such injuries among children and identifies the factors influencing their occurrence. The insights provided are crucial for enhancing treatment approaches and developing preventive strategies. This information is particularly relevant for shaping policies and guidelines related to dental trauma. I have reviewed the article and propose the recommendations and suggestions as listed.

Thank you.

Annotated reviews are not available for download in order to protect the identity of reviewers who chose to remain anonymous.

---

## Round 0.2 · Minor Revisions

Dear authors

We believe your manuscript would benefit from the remaining minor revisions suggested by one of the reviewers.

Reviewer 1 ·

Basic reporting

No remarks.

Experimental design

No remarks.

Validity of the findings

No remarks.

Additional comments

Dear authors,

in my opinion, you have significantly enhanced the quality of the manuscript by implementing the suggestions from the first round of the review. Therefore, it should be accepted for publication.

Congrats and kind regards!

Reviewer 2 ·

Basic reporting

Dear Editorial Team and Authors,

I have reviewed the original study titled "Traumatic Dental Injuries in Permanent Teeth Among Arab Children: Prevalence, Trends, and Associated Risk Factors – A Systematic Review and Meta-Analysis." This study provides a thorough examination of cross-sectional research conducted in Arab countries, highlighting the prevalence of traumatic dental injuries (TDI) in permanent teeth.

I recommend a few minor improvements to enhance the article.

Thank you

Experimental design

Abstract Section:
• Purpose: “The purpose of this study was to systematically assess the prevalence, trends, and potential risk factors of traumatic dental injury (TDI) in permanent teeth among children and adolescents in Arab countries.” The abstract should accurately reflect the results related to trends, which currently are not clearly presented.
o Prevalence: Yes
o Trends: The abstract should clarify the direction of trends, including changes in the incidence of traumatic injuries, behavioral patterns over time, or types of injuries.
o Risk Factors: Yes
• Conclusion:
o “Significant correlations were found between the prevalence of dental trauma and variables such as male gender, increased overjet, and inadequate lip coverage.” The result concerning lip protection is not mentioned in the abstract.
o “The prevalence of TDI in children and adolescents in the Arab region appears to be rising over time.” This statement “ appear to be rising over time” does not clearly reflect the findings presented in the abstract and needs to be better supported by the results.

Validity of the findings

Discussion Section:
• Consider adding a discussion on the association between overjet as a risk factor and related protective measures or interventions for Arab children. This should align with the research rationale, aiming to inform clinical practice, guide policy decisions, and identify areas for further research.

---

## Round 0.3 · accepted · Accept

Dear authors,
I would like to congratulate you on the substantial improvements made to the manuscript following the initial review. Your revisions have effectively addressed the previous concerns, enhancing both the clarity and overall quality of the work. Given these improvements, I am pleased to recommend your manuscript for acceptance without the need for further revisions.
Well done, and best regards.

Reviewer 1 ·

Basic reporting

No comment.

Experimental design

No comment.

Validity of the findings

No comment.

Additional comments

Dear Authors,

As mentioned earlier, in my opinion, your manuscript should be accepted for publication. However, after reviewing your manuscript once again, I have one suggestion that I would like you to consider before it is accepted.
Patients who experience dental fear and anxiety, regardless of age, present unique challenges that should be carefully addressed during the initial assessment and treatment planning for traumatic dental injuries. This consideration ensures the best possible care and overall experience for the affected patients. Moreover, the timely and appropriate treatment of traumatized teeth is a critical factor that significantly impacts their prognosis.
The aforamentioned should be taken into consideration as a limitation of the study, as these factors (dental fear and anxiety, the timely and appropriate treatment) have not been evaluated. Acknowledging this limitation will further strengthen the overall conclusions of your work.

Kind regards and congrats once again!